# Outcome differences by sex in oncology clinical trials

Ashwin V. Kammula [1], Alejandro A. Schäffer [1] ✉, Padma Sheila Rajagopal[1,2], Razelle Kurzrock [3] & Eytan Ruppin [1] ✉

Identifying sex differences in outcomes and toxicity between males and females in oncology clinical trials is important and has also been mandated by National Institutes of Health policies. Here we analyze the Trialtrove database, finding that, strikingly, only 472/89,221 oncology clinical trials (0.5%) had curated post-treatment sex comparisons. Among 288 trials with comparisons of survival, outcome, or response, 16% report males having statistically significant better survival outcome or response, while 42% reported significantly better survival outcome or response for females. The strongest differences are in trials of EGFR inhibitors in lung cancer and rituximab in non-Hodgkin's lymphoma (both favoring females). Among 44 trials with side effect comparisons, more trials report significantly lesser side effects in males ($N = 22$) than in females ($N = 13$). Thus, while statistical comparisons between sexes in oncology trials are rarely reported, important differences in outcome and toxicity exist. These considerable outcome and toxicity differences highlight the need for reporting sex differences more thoroughly going forward.

United States Public Health Service Act sec. 492B, 42 U.S.C. sec. 289a-2 states that "the Director of NIH [National Institutes of Health] shall ensure that the trial is designed and carried out in a manner sufficient to provide for a valid analysis of whether the variables being studied in the trial affect women or members of minority groups, as the case may be, differently than other subjects in the trial." The importance of analyzing and comparing males and females was strengthened as of 2016 when NIH instituted a policy to require analysis of sex as a biological variable (SABV) in preclinical studies[1]. In Janine Clayton's description of the pre-clinical policy implementation, the importance of following the law to compare males and females in clinical trials was re-emphasized[2].

One reason that studying sex differences in clinical drug trials is important is because physiological and immunological differences between males and females may dictate distinctions in drug behavior[3–8]. For instance, body composition and metabolism differences between the sexes might influence pharmacokinetic and pharmacodynamics of drugs[9]. There are sex differences in incidence rates

of many diseases including diabetes, cardiovascular diseases and cancer[5–8,10,11]. Furthermore, social constructs such as support systems might also show disparities by sex and could influence outcomes[12].

However, disappointingly, the law and the policy are not widely followed[13,14] and there is a lack of understanding that the requirement is to compare by (biological) sex and not by (patient reported identity) gender[15,16]. For example, the latest in a series of medium-scale meta-analyses of NIH-funded trials, reported in a paper published in a leading journal in 2015, found that only 26/107 "reported at least one outcome by sex or explicitly included sex as a covariate in statistical analysis"[13]. To consider at least one outcome by sex as a positive is a weak standard and does not meet the legal requirement that there should be a comparison by sex. If a study does a "subgroup analysis" of males and females separately, that does not actually offer a comparison.

Designing, analyzing, and evaluating clinical trials to examine and compare treatments in both sexes requires careful planning. It is important to recruit sufficiently many males and females to have some

[1]Cancer Data Science Laboratory, Center for Cancer Research, National Cancer Institute, Bethesda, MD 20892, USA. [2]Women's Malignancies Branch, Center for Cancer Research, National Cancer Institute, Bethesda, MD 20892, USA. [3]WIN Consortium and Medical College of Wisconsin, Milwaukee, WI 53226 and University of Nebraska, Omaha, NE 68198, USA. ✉e-mail: alejandro.schaffer@nih.gov; eytan.ruppin@nih.gov

power to detect differences[17]. Regulators must be clear and consistent about what sex-specific subgroup analyses and sex comparison analyses are expected in filings reporting trial results[18]. Differences in proper doses between males and females should be planned and should also consider the patient's age and pharmacogenomic markers[4,8,11,19]. Differences in death rates and aging patterns should be considered when comparing survival characteristics of males and females, especially when studying diseases, such as cancer, that predominantly afflict older patients[20–23].

Previous findings about differences in sex by survival, response and other outcomes are varied[14,24–26]. For example, there has been much debate about whether males have better outcomes in response to immunotherapy[27–33]. Others have hypothesized that females may have better survival in oncology clinical trials given their better survival in real-world data[10,20,34,35]. A very recent meta-analysis of oncology trials leading to drug approvals in the USA found no general significant differences in outcomes between males and females, even though individual trials may have significant differences[26]. In contrast, most studies and reviews on differences in side effects and toxicities by sex claim that females in oncology clinical trials experience more drug toxicities and other adverse events than males[3,36–41]. However, one recent large study reached the opposite conclusion[42].

Given the complex and unclear state of sex disparities in clinical cancer research, the purpose of this study is to comprehensively characterize sex outcome comparisons in all oncology interventional clinical trials and to identify those comparisons that find a significant difference between males and females. We aimed to include all interventional oncology clinical trials and research the following questions:

1. What outcome comparisons by sex have been reported according to cancer type, treatment, and measurement (e.g., survival or side effects)?
2. What types of evidence for sex differences are reported and how often?
3. For any recurrent patterns of outcome differences that we find, are the patterns already known and generally accepted based on previous meta-analyses or other study designs?
4. While reaching the expected conclusion that few sex comparisons are done, can we find any technical barriers to increasing adherence to the law and suggest ways to possibly overcome them?

In this work, we report three main findings based on a systematic curation and evaluation strategy across all oncology clinical trials by leveraging a paid service called Trialtrove that collects and curates data from ClinicalTrials.gov and thousands of other sources into a semi-structured format[43]. First, direct statistical comparisons by sex in outcomes or side effects in clinical trials results papers are rare. Second, females demonstrate better survival outcomes and treatment responses in the majority of clinical trials that do perform sex-specific statistical comparisons and identify a difference. Finally, we find that that there are marked sex-specific differences for particular treatments, namely epidermal growth factor receptor (EGFR) inhibitors in non-small cell lung cancer (NSCLC) and rituximab in non-Hodgkin's lymphoma (NHL), that extend beyond the overall underlying sex differences in survival in these two malignancies.

## Results
### Search, curation and comparison to ClinicalTrials.gov
We queried the 89,221 oncology clinical trials in Trialtrove as of December 23, 2022. The ~60% of trials with exact accrual data show an aggregate total of more than 6 million patients in our analysis. Subsequent curation found at least one sex comparison reported between males and females in 472 trials (0.5%). The general plan of curation and analysis is summarized in (Fig. 1a). Within this set of 472 trials, we identified 532 different post-treatment comparisons. 356 (66.9%) of these comparisons showed differences between males and females,

and 176 (33.1%) showed similarities (Supplementary Table 1). Each comparison was labeled as either being in *survival, outcome, or response (SOR)* or in a post-treatment *side effect (SE)* and was classified by strength of evidence provided (Fig. 1b).

In general, Trialtrove contains substantial information absent from ClinicalTrials.gov and we quantified this with one analysis. We evaluated on September 12, 2023 all 316 trials for which we found at least one sex comparison with a difference (not necessarily statistically significant). Among these 316 trials: 90 are not in ClinicalTrials.gov at all because they are in registries outside the USA, 142 are in ClinicalTrials.gov without any results, 79 mention sex only with respect to enrollment, which is called Participant Flow, and only 5 mention sex anywhere other than Participant Flow. Only 1 of these (NCT00418886) has separate analysis by sex. Ironically, the clearest indication that ClinicalTrials.gov does not record sex comparisons is trial NCT01274338 in which the associated paper entitled "Enhanced immune activation within the tumor microenvironment and circulation of female high-risk melanoma patients and improved survival with adjuvant CTLA4 blockade compared to males"[44] describes the sex comparison in the title; this paper is listed in the publications associated with trial NCT01274338. However, the Results subsection of the ClinicalTrials.gov entry for NCT01274338 has no analysis by sex. We recognize that ClinicalTrials.gov has neither the staff to do the curation nor the enforcement powers to require clinical trialists to deposit their results, so the Trialtrove curation adds considerable information.

### Overall sex differences in Survival, Outcome, Response (SOR) across clinical trials
Overall, we found 288 trials with statistical analysis of SOR outcomes. This set contained 47 (16.3%) trials with significant evidence favoring males (25S, 11O, 11R), 122 (42.4%) trials with significant evidence favoring females (89S, 11O, 22R), and 119 (41.3%) trials with statistical evidence showing no difference between males and females (71S, 7O, 41R) (Fig. 2a).

We analyzed these 288 trials by the treatments used in each (Fig. 2b, Supplementary Data 1). Most drugs with more than one trial (54/81, 66.7%) showed improved SOR in females (Fig. 2c). The three treatments most frequently assessed with sex-specific SOR analyses were erlotinib, gefitinib, and rituximab. For each of these treatments, we observed a statistically significant preference towards improved SOR in females relative to males (binomial test, FDR = 0.006, FDR = 0.011, FDR = 0.012 respectively).

We performed a binomial test for each category of treatment in our annotation (Methods). The Targeted (FDR = $8.5*10^{-5}$), Chemotherapy (0.004), and Antibody (0.01) groups all showed significantly more trials with improved SOR in females than males (Fig. 2d, Supplementary Table 2).

### Limited information about ages
Our primary analyses did not consider patient ages, although there are known interactions between sex and age affecting response to treatment[8,11,19,21,23,45]. Trialtrove coarsely annotates the ages eligible for a trial as any subset of three values, Children (ages 0–17), Adults (18–64), Older Adults (65–) and 71,832/89,221 trials have such an annotation. Among the oncology trials with an age annotation, the vast majority 59,792/71,832 (83%) have the age annotation "Adults; Older Adults". Among the trials with a sex comparison and an age annotation, the proportion annotated as "Adults; Older Adults" is similar at 294/362 (81%). The proportion of all trials annotated as exclusively "Older Adults" is only 1261/59782 (2.1%), precluding any further analysis of significance into the differences between adults and older adults.

### Quality control
Our analysis relies on Trialtrove curation, which could miss sex comparisons. Therefore, after doing most of our search curation, we

 

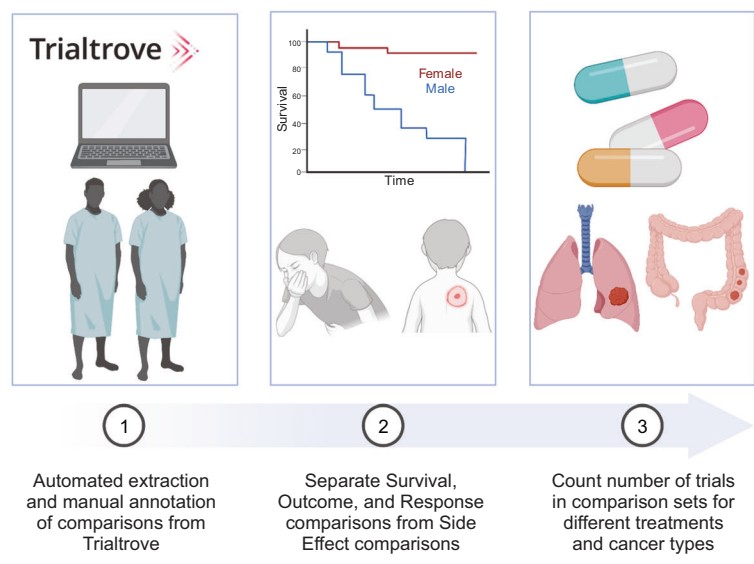

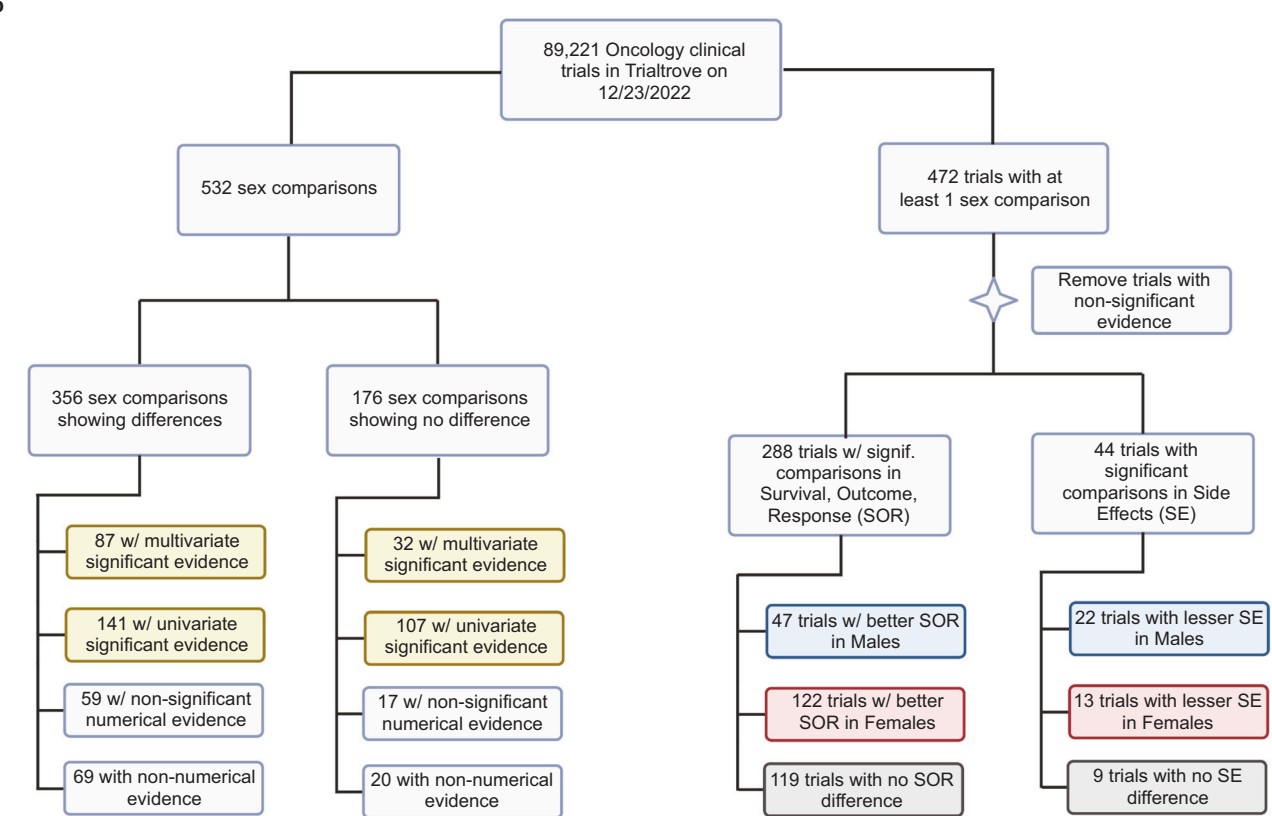

**Fig. 1 | Study design and summary result. a** Schematic of our study design. Subpanel 1) Using text mining methods we identified trials in Trialtrove that may have a comparison of outcomes or side-effects by sex. Subpanel 2) We individually curated each candidate trial to record what was measured such as a survival difference or a difference in the frequency of a side effect such as rashes; we recorded whether there were differences or similarities between the sexes and what evidence was provided. **b** A flow diagram showing the breakdown of data collected. All data points originated from the 89,221 oncology clinical trials in Trialtrove on December 23, 2022. On the left we show all 532 sex comparisons found, among which 356 showed sex differences and 176 showed no difference. These comparison sets are classified by the type of evidence they present. Yellow boxes indicate categories with significant evidence. On the right we show another view of the data flow centered on the trials for which we found sex comparisons (shown on the left). Filtering for trials which present statistically significant evidence, we show the 288 trials with SOR comparisons and the 44 trials with SE comparisons. These two groups are broken down further by whether they show a preference towards males (blue), females (red), or neither (gray). Panel **b** may be interpreted as a Preferred Reporting Items for Systematic Reviews and Meta-Analyses (PRISMA) flow diagram in which the unit of analysis is a clinical trial.

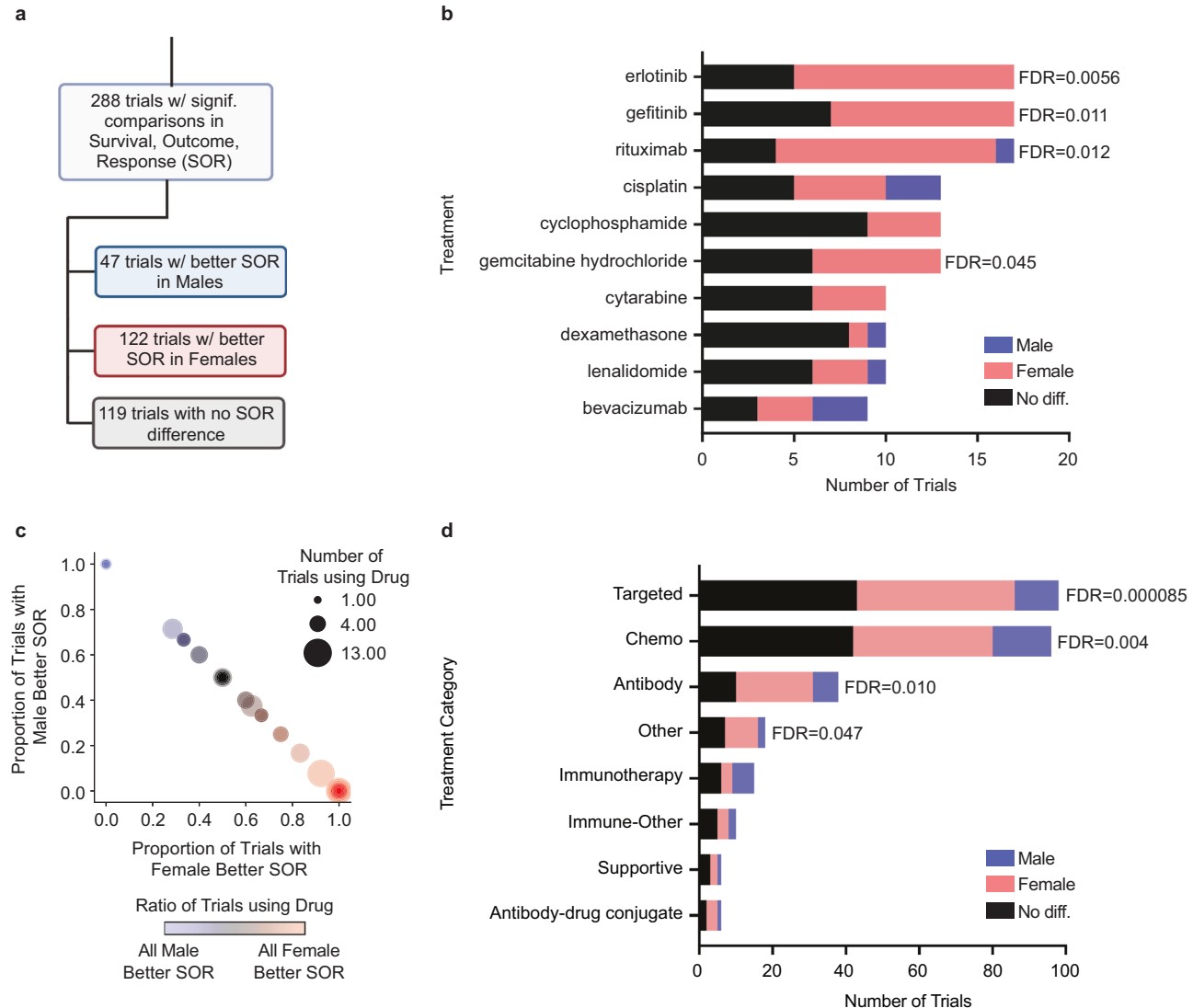

**Fig. 2 | Classification of trials with statistical survival, outcome, or response (SOR) comparisons by sex and by treatment. a** Counts of trials that have sex comparisons of survival, outcome or response (SOR) performed either via multivariate or univariate analysis; **b** Counts for the subset of trials from panel **a** that use any of the 10 treatments that have the most sex comparisons, with ties broken arbitrarily. Color indicates the sex with better SOR; **c** Scatterplot of 81 treatments that have more than one trial with a sex comparison, such that the x value is the proportion of trials favoring females and the y value is the proportion of trials favoring males; the size of each circle is the number of found trials for that treatment; the color of each circle is the proportion of trials favoring males; trials with no SOR sex differences are not shown in panel **c**; **d** Counts of trials from panel A according to the category of treatment given; treatment categories were ranked as

1. Immunotherapy, 2. Antibody, 3. Antibody-Drug Conjugate, 4. Targeted, 5. Chemo(therapy), 6. Immune-Other, 7. Other and 8. Supportive; a trial using multiple treatments of different categories was assigned to the lowest-number category of any treatment in the trial. In panels **b** and **d**, color indicates whether these trials show that males have better SOR (blue), females (pink), or that there is no difference (gray). Two-sided binomial tests were used to determine whether the proportions of male-favoring to female-favoring trials were significantly different from 1:1. *P*-values were corrected for multiple hypothesis testing using the Benjamini–Hochberg method. Significant FDR values are displayed in the figure. All others are available in Supplementary Tables. Source data are provided as a Source Data file.

selected 75 trials with large enrollments ≥200 including both males and females that appeared to have Trialtrove-curated results but did not have a sex comparison identified by our analysis (see Methods subsection entitled Quality Control of Query and Preliminary Results). These 75 trials are among the largest oncology trials done and hence are likely to have sufficient statistical power to detect a sex difference in outcomes if one exists. We intentionally did this quality control analysis shortly before finishing our curation so that if we found possible improvements in our methods, we could implement those improvements and we did implement one improvement.

To this end, we searched in detail any papers and abstracts published about those 75 large trials to see if any sex comparisons were missed by Trialtrove curators. The results of our quality control analysis

are in Table 1. The large majority of trials 65/75 (87%) of trials had a published paper by November 2022, so we could realistically assess whether the authors did an analysis by sex; for the other 10/75 (13%), the results in Trialtrove are based on conference abstracts or other brief communications. More than half the trials 38/75 (51%) had a paper with no analysis by sex confirming the concern that led to our study. For only 1 of 75 large trials we checked, there was a statistically significant sex comparison in the main document of the publication that Trialtrove curators missed. We infer that Trialtrove curators found the large majority of statistically significant published sex comparisons. As expected, there was a larger number of trials (8/75) that had an insignificant (7/75) or marginally significant (1/75, whether it is significant depends on not correcting for multiple tests) sex comparison relegated

**Table 1 | After completing an initial attempt to find trials with sex differences based on an April 2022 data freeze, we carried out various quality control tests (Methods)**

| Reason trial was missed | Number among 75 selected trials with enrollment ≥200 | Number among 147 trials with enrollment [175, 199] |
|---|---|---|
| Paper exists but lacks any analysis by sex | 38 (51%) | 95 (65%) |
| Paper has subgroup analysis by sex but no comparison | 17 (23%) | 10 (7%) |
| No paper as of end of evaluation | 10 (13%) | 23 (16%) |
| Paper has non-significant comparison missed by Trialtrove | 7 (9%) | 17 (12%) with 2/17 papers published after data freeze. |
| Sex comparison in Trialtrove missed by our strategy, subsequently repaired | 1 (1%) | 0 (0%) |
| Paper has a significant SOR or SE sex comparison in main document missed by Trialtrove | 1 (1%) | 2 (1%) |

In one key test, we examined 75 larger trials likely to have a sex comparison that were missed by our search of the April 2022 data. In the middle column, we classified each trial into one of seven categories. The middle column may include papers curated by Trialtrove between the April and December 23, 2022 data freezes. The end of evaluation date for the right column was September 23, 2023. For the trials with no paper, Trialtrove curation could have found sex comparisons in meeting abstracts, without a peer-reviewed journal paper, but did not actually find any sex comparisons in abstracts.

to the supplementary information; hence, Trialtrove curators missed the sex comparisons in those 8/75 publications. For the other 66/75 trials, no sex comparison was published. The most surprising and important finding is that a large proportion of papers (17/75, 23%) had a subgroup analysis by sex in which males and females were analyzed separately but no comparison was done. In the next paragraph and in the Discussion, we hypothesize as to why this practice of subgroup analysis of males and females separately with no sex comparison has arisen.

Subgroup analyses were always presented as a forest plot that included a row for the hazard ratio for males and a row for the hazard ratio for females; usually the hazard ratio compares two arms (e.g., new treatment vs. standard of care) for some form of survival or response. Most (14/17) of the papers that did separate analyses of males and females are published in one of four journals: Journal of Clinical Oncology (4), Lancet (3), Lancet Oncology (1), New England Journal of Medicine (6). In each of these journals the Editorial Office replots the forest plots for subgroup analysis in a nearly homogeneous format. One of the few inhomogeneities is that some of these forest plots include an accompanying direct comparison of male and female outcome, for example by a Cox regression test of survival, but most do not include such a comparison.

The observation that Trialtrove curation missed 8 comparisons that were not statistically significant and/or relegated to a supplement quantifies the curation bias against non-significant results and helps explain why we found so many fewer comparisons with similarities (176) than with differences (356) between sexes. Therefore, we focused most of our downstream analyses on the statistically significant comparisons. Since Trialtrove curation missed only one significant comparison in the main document, we suggest that our collection of significant comparisons is representative of the available comparisons.

At the suggestion of a reviewer, we added a second, larger assessment of all 147 trials that were eligible for our main analysis and were not found by us to have a sex comparison curated in Trialtrove, appeared to have results, and had enrollments in the slightly smaller range of [175, 199]. The enrollment criterion was selected to prefer large trials that have power to find sex differences while avoiding any overlap with the first assessment, which required enrollment ≥200. Encouragingly, we found only 2/147 trials with a significant SOR or side effect sex comparison that Trialtrove curators missed and 0/147 that our search methods missed (Table 1). The main differences in the outcomes of the two quality control assessments were an increase in the proportion of studies with a paper but no analysis by sex (51% in the first assessment and 65% in the second assessment) and a corresponding decrease (23% in the first assessment to 7% in the second assessment) in the studies with a paper that did a separate assessment of males and females. Possible reasons for this difference include that

i) larger trials are more likely to have power to analyze males and females separately and ii) larger trials are more likely to be published in very high impact journals such as New England Journal of Medicine and Journal of Clinical Oncology, which have developed standardized, in-house figure designs for forest plots that are used to illustrate subgroup analyses, such as separate analyses of males and females. The trials with enrollment <200 are naturally less likely to be published in the highest impact journals and the lower impact journals do not necessarily encourage authors to do analyses by sex, either separately or in comparison (see Discussion).

## Analysis of trials by starting year

To see how the use of sex comparisons has changed over time, we calculated the proportion of candidate trials that report sex-specific subgroup analyses as a function of the starting year of the trial. The proportion of trials with sex-specific comparisons decreased over time in trials of all phases, and this finding was consistent when trials were further split to Phase II and Phase III only (Supplementary Fig. 1, Methods, Supplementary Table 3). The decline was especially substantial between trials with starting years in 2008–2009 (58/2030 trials had sex comparisons) and trials with starting years in 2016–2017 (20/1985 trials). This difference in proportions is highly significant ($P < 4E-5$, Fisher's exact test, two-sided). We suggest that there are two overlapping reasons for this decline. First, as explained in the Discussion, in the first decade of the 21st century caution among biostatisticians about subgroup comparisons increased, especially regarding post hoc subgroup comparisons. Second, as we observed above, recent trial papers tend to report separate analysis of males and females as well as other subgroups but tend to avoid direct comparisons between subgroups. We provide more fine-grained data by single year and for each trial phase in Supplementary Data 2. As an influential anecdotal example, we mention a meta-analysis of sex and response to immune checkpoint blockade; all 23 of the underlying trials analyzed males and females separately without a direct comparison[32]. Of note, we found the highest fraction of candidate trials with sex comparison in the Phase III set (Supplementary Fig. 1), suggesting that as drugs near requests for regulatory approval, doing comparisons by sex increases in importance.

It may be argued that the denominator of 89,221 is an overestimate for several reasons. Therefore, we counted the subset of trials that had an enrollment of more than 25, enrolled both males and females, had a known start year of 1993–2022, and had at least one data collection location in the United States including Puerto Rico. The last two requirements are because the United States Public Health Service Act sec. 492B, 42 U.S.C. sec. 289a-2 (quoted at the start of the Introduction) was enacted only in the United States in 1993. The number of trials in the numerator and denominator meeting the four requirements above are 215/17988 (1.2%). The numbers for each start

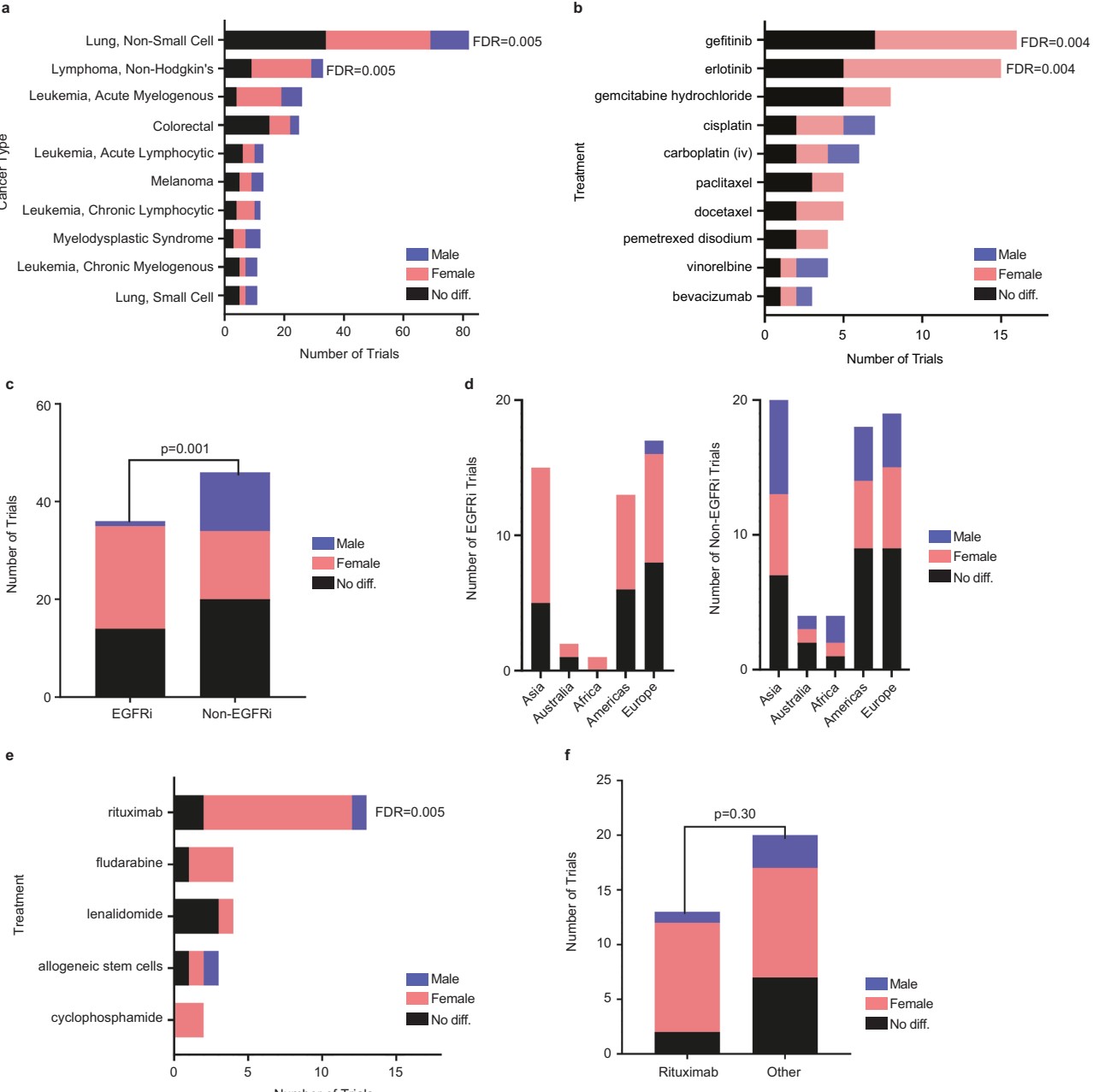

**Fig. 3 | Classification of trials with statistical survival, outcome, or response (SOR) comparisons by malignancy with a focus on non-small cell lung cancer (NSCLC) and non-Hodgkin's lymphoma (NHL). a** Counts of the trials from Fig. 2a by malignancy for the 10 malignancies with the most sex comparisons of SOR; **b** counts of NSCLC trials from **a** according to the treatment given; **c** Analysis of NSCLC SOR trials from **a**, split by those which used an EGFR inhibitor (EGFRi) and those which did not use an EGFR inhibitor (non-EGFRi). **d** Classification of NSCLC trials with statistical SOR comparisons by continent; trials on multiple continents were counted once for each continent; no continents have an over-representation of or under-representation of any continent among trials favoring females. **e** Classification of non-Hodgkin's lymphoma (NHL) for the five most common treatments in trials with an SOR sex comparison. **f** Analysis of NHL SOR trials from **a**, split by those which used rituximab and those which used 'Other' non-rituximab treatments. In all panels, color indicates whether these trials show that males have better SOR (blue), females (pink), or that there is no difference (dark gray). Figures **a**, **b** and **e** used two-sided binomial tests to determine whether the proportion of male to female trials was significantly different from 1:1. *P*-values were corrected for multiple hypothesis testing using the Benjamini–Hochberg method. Significant FDR values are displayed in the figure. All others are available in Supplementary Tables. Figures **c** and **f** utilized two-sided hypergeometric tests to assess whether EGFR inhibitor trials or rituximab trials were enriched for improvement in females. Significant FDR values are displayed in the figure. All others are available in Supplementary Tables. Source Data are provided as a Source Data file.

year are shown in Supplementary Table 4. To summarize the analysis over time, we do not observe any increase in the proportion of United States -based trials with a sex comparison after any of four key events: enactment in 1993 of the law mentioned above, establishment of ClinicalTrials.gov in 2000, requirement of trial registration in ClinicalTrials.gov starting around 2007, and implementation of the NIH policy on sex as a biological variable around 2016.

## Two treatment/cancer type combinations drive favorable SOR outcomes for females: EGFR inhibitors in NSCLC and rituximab in NHL

We further analyzed trials with significant SOR comparisons by cancer type (Fig. 3a, Supplementary Table 5). NSCLC had the largest number of trials with systematic statistical SOR comparisons, 82 in total. There were significantly more trials with SOR that favored females (*N* = 35,

42.7%) than trials with SOR favoring males ($N = 13$, 15.9%) (FDR = 0.005). 34 (41.5%) trials showed no SOR difference between males and females.

We analyzed the NSCLC trials by the treatments used in the comparisons (Fig. 3b, Supplementary Table 6). We found that 36/82 NSCLC trials used epidermal growth factor receptor inhibitor (EGFRi) treatments, predominantly gefitinib and erlotinib. Of the EGFRi trials (Methods), strikingly, 21 (58.3%) had SOR favoring females, 1 (2.8%) favored males, and 14 (38.9%) observed no difference. In contrast, among all NSCLC trials involving other treatments, the emerging picture is more balanced, with 14 (30.4%) studies showing favorable SOR in females, and 12 (26.1%) studies favoring males (Fig. 3c). The EGFR inhibitor trials have preferential SOR in females which goes beyond the overall finding that females have better SOR than males with NSCLC ($P = 0.001$, hypergeometric test, Methods). Given the hypotheses of potential sex-specific differences based on a higher frequency of somatic *EGFR* mutations among NSCLC tumors from non-smoking females in Asia, we evaluated sex-specific difference reporting by continent and did not observe a unique pattern relative to other continents (Fig. 3d, Supplementary Table 7).

In NHL, we identified 33 trials with significant SOR comparisons, 20 (60.6%) showing SOR that favored females, 4 (12.1%) with SOR favoring males, and 9 (27.3%) trials showing no SOR difference (Fig. 3e). Notably, of 13 trials that used rituximab, 10 showed better SOR in females and only 1 in males, with 2 trials showing no significant difference (Fig. 3f, Supplementary Table 8). Given that rituximab is used in other non-cancer contexts, we sought to assess if there is a sex-specific association with rituximab across other types of trials. In Trialtrove, we identified 48 such trials, but including only 4 sex comparisons, reporting balanced results (Methods). Comparing the distributions of NHL trials that used rituximab to those which did not, the difference between the two is not significant ($P = 0.30$).

Acute myelogenous leukemia (AML) was the malignancy with the third largest difference between number of trials with SOR results favoring females and males, but not in a statistically significant manner (FDR = 0.2). These AML trials used a variety of different treatments, and we could not detect any related treatment specific patterns. Thus, AML may be a malignancy in which the better SOR in females manifests across different treatments and clinical trials, in difference from NSCLC and NHL

### Analyses that would be interesting but cannot be done with Trialtrove data

The interaction effect between sex and age on morbidity and mortality is well established, with women living longer and experiencing greater frailty in older age. We are not able to observe the effect of this interaction with the available data[21,23,45,46]. Similarly, pharmacology studies have demonstrated that pharmacokinetics and pharmacodynamics are different between males and females[4–6,8,9,11,19,46], with these factors as contributors for women experiencing more side effects (potentially older women more specifically). Regrettably, only 33 of the 472 trials in Trialtrove that performed sex-specific comparisons describe any data collection related to any of the cytochrome P450 genes involved in drug metabolism, precluding any systematic analysis of such interactions.

### Males have fewer side effects than females

Overall, 97 (18.2%) of all comparisons we found were regarding post-treatment side effects (SE), often in trials of drugs intended to mitigate side effects. SE comparisons with multivariate or univariate evidence were performed in 44 trials. Those 44 include 22 trials (50%) that showed statistically significant lesser side effects in males, 13 (29.5%) in females, and 9 (20.5%) with no significant difference (Fig. 4a). These trials are reported by treatment, treatment category, and cancer indication (Fig. 4b, Supplementary Tables 9 and 10, Supplementary

Data 3). Among these 44 trials, colorectal cancer is the most prevalent indication, with 15 trials overall, 10 (66.7%) favoring males (that is, lesser side effects in males), 3 (20.0%) favoring females, and 2 (13.3%) reporting no difference. Notably, 6 of these colorectal cancer trials used oxaliplatin, 5 (83.3%) favoring males and 1 (16.7%) favoring females (Supplementary Table 11).

## Discussion

We searched all oncology clinical trials in Trialtrove to identify trials that compared survival, outcome, or response (SOR) or side effects (SE) between males and females. This approach allowed our study to explore across cancer types and treatments to an extent that is orders of magnitude larger than prior studies. We reviewed the 89,221 oncology trials reported in Trialtrove between 1974 to 2022. We report three main findings: First, direct comparisons by sex in SOR or SE visible enough in clinical trial results papers to be curated by Trialtrove are rare−only 0.5% of studies included such a comparison. Second, females have better post-treatment SOR in most clinical trials that performed sex-specific comparisons. This difference is largely driven by trials in NSCLC, NHL, and AML rather than across all cancer types. Despite the better SOR outcomes for females versus males, though, males have significantly less toxicity than females in the subset of trials that could be analyzed for that parameter. Third, sex-specific differences are marked for particular therapies as applied to specific cancer types, namely EGFR inhibitors in NSCLC and rituximab in NHL, beyond the underlying overall sex differences observed in these two cancer types.

In response to our second research question regarding types of evidence, we found only 472 of 89,221 trials that reported performing curated sex-specific comparisons (a total of 532 such comparisons were done). This result is striking and discouraging given existing U.S. law and NIH goals. A common reason we observed is that clinical trials tend to report subgroup analyses, where males and females are analyzed separately, but not directly compared (Table 1). In clinical trial biostatistics, the common practice of analyzing males and females separately against the whole trial population can be improved by adding a Cox regression analysis, a statistical test of interaction, and/or a correction for testing of multiple subgroups[47], to compare survival between males and females. Sun and colleagues describe an influential set of 11 subgroup analysis rules that should be considered in this context[48]. These rules are well-intentioned and appear to have improved data analysis practice over the past 10 years[49]. One of the rules is that the direction the subgroup effect should ideally be specified a priori. Our results support the need for a priori specification of subgroup testing by sex and potentially offer direction and estimation of subgroup-specific effects for clinical trial development. Another reason that we may not have observed sex-specific comparisons is that non-significant comparisons may not have been discussed or only shown in the supplementary information (which Trialtrove does not curate).

With regard to our third research question and identification of existing sex-specific patterns, the sex-specific association between EGFRi and NSCLC has been previously characterized in several studies[10,50–53] and has been attributed to a lower proportion of female smokers vs. male smokers and a higher proportion of female patients with EGFR somatic mutations vs. male patients[54,55]. EGFR mutations are known to be more prevalent in Asian countries and this epidemiological distinction is thought to explain why early trials of EGFR inhibitors, which did not consider EGFR mutation status, were more successful in Asia compared to other continents[54]. However, our geographic analysis of EGFRi trials does not support the hypothesis that the female advantage in EGFRi NSCLC trials rests on trials from Asia.

We also found that females with NHL have better SOR than males when treated with rituximab. Previous reports of improved SOR for females in NHL/rituximab were generated from single trials or

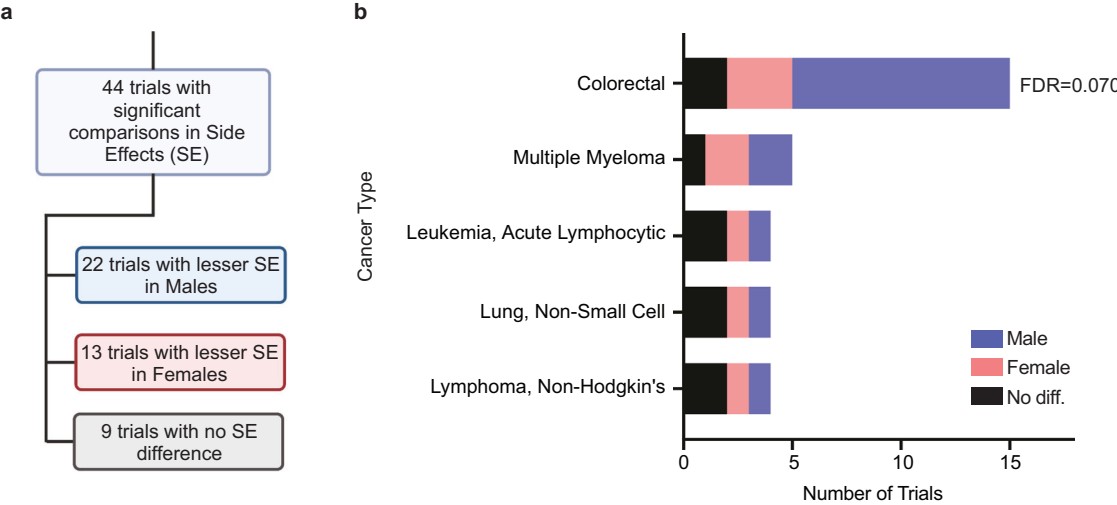

**Fig. 4 | Analysis of trials with statistical comparisons of side effects (SE). a** The numbers of trials with a multivariate or univariate comparison of side effects between males and females. lesser SE in Males means here that the side effect is statistically less common or less severe in males. **b** Counts of the trials in panel **a** for the five malignancies with the most comparisons - FDR for Colorectal cancer is not far from significant, *P* = 0.07. For each malignancy, a two-sided binomial test was used to determine whether the proportion of male-favoring (blue) to female-favoring (pink) trials was significantly different from 1:1. *P*-values were corrected for multiple hypothesis testing using the Benjamini–Hochberg method. Significant FDR values are displayed in the figure. All others are available in Supplementary Tables. Source data are provided as a Source Data file.

anecdotal evidence collected from a few trials[56], but the sex-preference in response to rituximab has not been systematically reported across trials. Importantly, other anecdotal reports of sex differences for various malignancy/treatment combinations[25,31,57,58] are not supported by our analysis of all oncology trials. Notably, rituximab does not show consistently better outcomes in females across other B-cell disorders[59], suggesting that oncology clinical trials may reflect an interaction of NHL and rituximab with sex specificity, rather than a sex-specific pharmacodynamic property[60]. Further mechanistic studies are needed to learn if these differences are a result of sex hormone interactions with the drug, differing mutation frequencies between males and females, or other effects.

There are, of course, limitations to this work. This analysis relies on Trialtrove curation to capture reported sex differences from published papers. Careful, formal languages analysis (Methods) of the Trialtrove text avoided a frequent problem that (biological) sex is confused with (self-identified) gender[7,16]. We manually curated the Trialtrove results as well, but trials where sex comparison information was not reported in Trialtrove would not have been reviewed, except as part of our quality control assessments. As we noted, non-significant sex comparisons may be reported solely in the supplementary information. Indeed, the fact that we found 356 comparisons that found sex differences compared to 176 comparisons that found similarities, testifies to the known publication bias against reporting non-significant results.

Our analysis showcases the impact from and room for improvement in current policies to identify sex-specific results in clinical trials. A 2014 US Food and Drug Administration (FDA) Action Plan (https://www.fda.gov/media/89307/download) highlighted 27 actions divided into the priorities of "improving completeness/quality of demographic subgroup data collection, reporting, and analysis; identifying barriers to subgroup enrollment in clinical trials and employing strategies to encourage greater participation; and making demographic subgroup data more available and transparent." Additionally, projects awarded by the FDA's Office of Women's Health Research will start to address some of the questions we bring up with this work, but this group has the funding for only a limited number of projects with a duration of 1–2 years per project (https://www.fda.gov/science-research/womens-health-research/list-owh-research-program-awards-funding-year#2024). Prioritization cannot be at the level of the FDA alone. Incentives

for recruiting sufficient patients and performing these comparisons must also be at the level of journals. As of 2016, several top-tier scientific and oncology-specific journals and journal families, including The Lancet family, Journal of the National Cancer Institute, the Cell family, the Nature family, and the Science family, have adopted SAGER guidelines that require reporting of sex/gender of participants and, to some extent, justification for inadequate powering for subgroup analysis[61,62]. SAGER guidelines or some similar alternative should be the norm across journals.

In conclusion, direct comparisons by sex in outcomes or side effects in papers reporting clinical trial results are still very rare. This is despite increasing interest in sex differences in clinical medicine and pharmacology[3,7,9,10,14,25,38,58,63,64], the requirement of US law and its implementation by the NIH[2,32,65,66], and interesting examples where different treatments by sex may lead to better collective outcomes[28,67]. Our findings of treatment-specific biases even in the current sparse comparisons supports the urgent need to perform sex comparisons on a much wider scale, which would likely reveal additional clinically important observations. Clinical trialists, biostatisticians and journal editors are in positions to highlight subgroup differences and to improve our understanding and leveraging of sex-specific treatment outcomes.

## Methods
### Trialtrove
We systematically searched Trialtrove (https://citeline.informa.com/trials/results), an online repository of clinical trials to collect a set of trials that identified a difference in a post-treatment outcome between male and female patients. Trialtrove reports and summarizes each trial in 75 semi-structured data fields. We elected to query Trialtrove, rather than the more commonly used ClinicalTrials.gov, because Trialtrove has more consistent formatting that allows for use of formal language methods[68]. Trialtrove includes more treatment intervention trials than ClinicalTrials.gov because Trialtrove collects data including and beyond ClinicalTrials.gov[43]. The full Trialtrove data are available only under license, so we can only provide summary information[69]. Recent large studies that similarly used Trialtrove include a study predicting drug approvals, a study about the use of germline information in clinical trials, and a catalog of immunotherapy trials[69–71].

Almost all results shown below are based on one consolidated search of a frozen and downloaded set of all 89,221 oncology trials present in the Trialtrove database on December 23, 2022. Data were processed during December 2022-March 2023. Usage of one earlier data freeze to explore search strategies and for quality control is described in the subsection below entitled Quality Control of Query and Preliminary Results. Trialtrove is updated every weekday and in our experience, new data in ClinicalTrials.gov appear in Trialtrove within days or a few weeks. To have a clear and consistent reference point, we had to take a single data freeze (in December 2022) to have a stable set of data to analyze.

## Two data freezes

We searched in two major phases. The first phase (June–November 2022) was based on a Trialtrove data freeze in April 2022 and was intended to optimize our search methods. We initially expected that that any sex comparisons would be found in the column named 'Trial Results' but quality control revealed that many sex comparisons are instead in the field 'Trial Notes'. Almost all results in this study are based on one consolidated search of a frozen and downloaded set of all 89,221 oncology trials present in the Trialtrove database on December 23, 2022.

## Database filtering

To filter the 89,221 trials, we placed the following restrictions: the trial must treat patients of both sexes (the "Patient Gender" column must be equal to "Both"), the trial must have at least 25 patients enrolled or must not have specified the number of patients (the "Actual Accrual (No. of patients)" entry must be greater than or equal to 25 or must be blank), and the trial must have results (the "Trial Results" or "Trial Notes" fields must contain of one of the below terms which suggest that results are reported).

['ORR', 'CR', 'DCR', 'RFS', 'OS', 'DFS', 'disease-free survival', 'LDFS', 'IDFS', 'PFS', 'progression-free survival', 'event-free survival', 'PFS4', 'FFP', 'Objective response', 'objective response', 'Complete control', 'complete control', 'Complete response', 'complete response', 'Overall response', 'overall response', 'Partial response', 'partial response', 'Disease control rate', 'disease control rate', 'Tumor response', 'tumor response', 'Survival rate', 'survival rate', 'Survival rates', 'survival rated', 'Response rate', 'response rate', 'Response rates', 'response rates', 'Remission rate', 'remission rate', 'Effective rate', 'effective rate', 'free survival was', 'pCR', 'PCR', 'mCR', 'nCR', 'cCR', 'QoL', 'pathological response', 'Pathological response', 'clinical response', 'Clinical response', 'cytogenetic response', 'Cytogenetic response', 'CCyR', 'hematological response', 'Hematological response', 'hematologic response', 'Hematologic response', 'CCR', 'recurrence rate', 'Recurrence rate', 'recurrence rates', 'Recurrence rates', 'CHR', 'cumulative response', 'distant metastasis-free survival', 'DMFS', 'durable clinical benefit rate', 'durable response rate', 'durable responses', 'Early molecular response', 'EMR', 'event-free survival', 'local failure-free survival', 'LFFS', 'major cytogenetic response', 'MCyR', 'major molecular response', 'MMR', 'mean duration of the response', 'Mean duration of the response', 'median treatment duration', 'Median treatment duration', 'median follow up', 'median followup', 'molecular response', 'MR', 'PCR rate', 'radiological response', 'regional failure-free survival', 'RFFS', 'resection rate', 'Resection rate']

## Identifying trials with sex comparisons

After this initial filtering, we narrowed the trial list to those in which we believed a comparison between males and females was reported in either Trial Results or Trial Notes. From the Python3 package re (v2.2.1), we used the findall() method with a regular expression of '\w+' to split the Trial Results or Trial Notes field into an array of tokens where each token represented a word in the associated context. We

required that one of these fields must contain a term regarding sex ("male", "female", "men", "women", "males", "females", "m", "f", "gender", "sex") within nine tokens of either:

1. A term suggesting a comparison: 'more', 'less', 'fewer', 'greater', 'higher', 'lower', 'frequent', 'frequently', 'preferential', 'preferentially', 'associated', 'similar', 'similarly', 'better', 'compare', 'compared', 'difference', 'greater', 'longer', 'odds', 'rate', 'response', 'responses', 'shorter', 'significant', 'significantly', 'statistical', 'statistically', 'versus', 'vs', 'worse', or 'worst'
2. A term that may represent an outcome or side effect. (terms listed in the section Database Filtering)

The steps up to this point identified 11,259 candidate contexts over 4061 trials. Many of these contexts were detected as false positives in a semi-automatic manner due to extraneous uses of the terms 'm' and 'f' representing something other than 'male' and 'female' (see the subsection entitled Examples of Why 'M' and 'F' Usually Do Not Represent 'Male' and 'Female'). Duplicate trials were removed. The remaining trials were manually curated as described in the next subsection, including checking original papers and abstracts if the Trialtrove annotations were ambiguous or incomplete.

## Curation and annotation

In the curation process, we classified each comparison as either being in survival, outcome, or response (SOR) or in a post-treatment side effect (SE). Those labels were determined in our curation based on the text involved in the comparison: The label 'Survival' represents comparisons which explicitly state survival. The 'Response' label stands for response outcomes reported via the RECIST criteria as well as measurements of time to progression and relapse. The label 'Outcomes' refers to other reports of outcome, such as improvements in quality of life, that are not a formal measurement of survival or of response. The 'SE' label refers to any post-treatment toxicity or side effect regardless of severity, including nausea and vomiting, anemia neutropenia, and rashes, among others. Each sex comparison was additionally annotated for the type of evidence provided in one of four ranked categories: Multivariate Analysis Significant, Univariate Analysis Significant, Other Numerical Comparison, No Numerical Comparison. Most downstream analyses were focused on the 288 SOR and 44 SE trials that presented statistically significant differences or similarities with statistical tests. Due to a small number of trials analyzing side effects by sex, we did not subset the side effect category by severity or type.

Following automated trial filtering, each trial was manually annotated by both A.V.K and A.A.S. To aid our manual annotation, we automatically stored the textual context with nine words on either side flanking the sex term. We looked beyond the context to more of the Trial Results or Trial Notes field if the context was not sufficiently clear. We also looked in the original sources cited in Trialtrove as needed to clarify ambiguities in the Trialtrove curation. In our annotation, we aimed to answer the following questions:

1. Is there a true comparison being made between males and females?
2. Is there is a difference or no difference between males and females?
3. Could the comparison have been done before treatment (even if it was done later)?
4. What patient measurement (e.g., survival) is being compared?
5. What is the evidence type used in the comparison?
6. If there is a difference, does the difference represent an improvement in males or females?
7. In what disease(s) was this comparison made?
8. Which treatments used (among the Primary Tested Drugs field) led to this comparison being made?

When annotating comparisons by type (Question 4), we place comparisons into one of four categories: Difference in Survival (comparisons that explicitly state survival, such as overall survival, progression-free survival and event-free survival), Difference in Response (as reported by RECIST criteria or other measurements of progression and relapse), Difference in Other Outcome (all other reports of outcome such as Health Related Quality of Life (HRQOL), presence of brain metastases, and relative risk of death), Difference in Side Effect (any post-treatment side effect or toxicity). Additionally, when annotating comparisons for evidence type used, the possible options are Multivariate Analysis, Univariate Analysis, Other Numerical Comparison, No Numerical Comparison. Using these categories, a trial has a comparison that is not different (shortened to no diff. in the Figures) if either:

A. The evidence type is Multivariate Analysis or Univariate Analysis and the comparison was not statistically significant or

B. The evidence type is Other Numerical Comparison or No Numerical Comparison and the curation described the comparison with an adjective such as "same" or "similar" or "nearly identical".

### Examples of why 'M' and 'F' usually do not represent 'male" and 'female'

When looking for terms that may represent one sex or the other, we included the single letter abbreviations 'M' and 'F' in either upper case or lower case. These initials do represent 'male' and 'female' in a miniscule percentage of Trialtrove entries, but usually they represent something else.

M or m can appear in author initial, short for "months", short for "meter", in M.D., and in "m protein" as an initial for a treatment such as "methotrexate"; F or f can appear in author initial as an abbreviation for a treatment such as "fluorouracil" and as part of an html (hypertext markup language) string used in Trialtrove syntax, the sixth arm in a trial with at least six arms a, b, c, d, e, f.

### Removal of candidate trials with duplicated information

To avoid double counting, we aimed to remove Trialtrove entries with overlapping results representing the same clinical trial or an umbrella trial including various sub-trials that have their own Trialtrove entries. Two trials were defined to be duplicates if the context of each (sex term, comparison term/results term) pair in one trial was contained in the contexts of the other trial. If the two trials had identical context sets, then the trial with the lower Trialtrove ID was removed. If one trial's context set was a subset of another, then the smaller context trial was removed.

### Statistics & reproducibility

Statistical tests were performed using the Python3 packages SciPy (v1.7.3) and statsmodels (v0.13.2). We performed subsequent analyses using Python3 and Pandas. We split our comparison set into a set of comparisons in survival, outcome, or response (SOR) and a set of comparisons in side effects (SE) and only considered comparisons with either multivariate or univariate significant evidence. For downstream analyses, rather than analyzing by comparisons, we analyzed by trials. This distinction is important because a trial may contain more than one comparison. Within the SOR comparison set, trials were classified based on the comparison with comparison type highest in this ranking; Survival > Response > Outcome.

Downstream analyses included classification according to the Disease field and the drugs tested. For one set of analyses combining different treatments, we (A. A. S. and S. P. R) classified treatments into the following mutually exclusive, ranked categories:

Immunotherapy (immune checkpoint blockade, 10)
Antibody (that are not immunotherapy, 22)
Antibody drug conjugate (3)
Targeted (71)
Chemotherapy (90)
Immune-Other (31)
Other (35)
Supportive (28)

The classification was done by expert knowledge and by using a published table of drugs to distinguish Targeted from Chemotherapy[72]. A trial was assigned to its highest ranked treatment category for the purpose of analyzing trials by these eight categories.

For both the SOR and SE comparisons, we counted the number of male, female, and same trials for different treatments, treatment categories, and cancer types. To determine whether a specific subset of trials was enriched for trials favoring males or females, we performed a two-sided binomial test of the hypothesis that the male favored to female favored trials are in a 1:1 proportion. Two-sided binomial tests were only performed if the number of male trials plus the number of female trials exceeded 3. Comparisons of three or fewer trials were deemed to have insufficient statistical power to draw meaningful conclusions. For each set of analyses where we divided the trial set by a specific variable (treatment, treatment category, cancer type), we used the Benjamini–Hochberg method to correct for multiple hypothesis testing and to calculate the false discovery rate (FDR). Binomial tests were performed using the binom.pmf() function from scipy.stats. FDR corrections were performed using the multitest.fdrcorrection() function from statsmodels.stats. Any analysis with FDR corrected $p$-value ≤ 0.05 was considered significant.

The above analysis was additionally performed by subsetting the non-small cell lung cancer trials and non-Hodgkin's lymphoma trials by treatment.

To determine whether non-small cell lung cancer trials using EGFR inhibitors (afatinib, cetuximab, erlotinib, gefitinib, vandetanib) had significantly more female improved trials than NSCLC trials not using EGFR inhibitors, a hypergeometric test was performed, using the hypergeom() function from scipy.stats. The same analysis was performed to determine whether non-Hodgkin's lymphoma trials using rituximab had significantly more female improved trials than NHL trials not using rituximab.

The principal sample size was determined by all available trials in Trialtrove that have curated sex comparisons. No trials were excluded. For analyses of trials of specific diseases (such as NSCLC) or specific treatments (such as rituximab) all oncology trials meeting those criteria were included. Since all applicable oncology clinical trials were included, no statistical method was used to predetermine the sample sizes. No data were excluded from the analyses. There was no randomization and no blinding. Curation of the trials in the Source data was done by A.V.K. and A.A.S who checked each other's work to arrive at a consensus curation for each trial.

### Quality control of query and preliminary results

In November 2022, using the April 2022 data download, we selected 75 trials that i) were not found to have a sex comparison, ii) appeared to have results based on the Trial Results field and iii) appeared to have enrolled at least 200 patients and patients of both sexes, making it likely that there was some power to detect a sex difference. We used the 75 trials selected in November 2022 to assess what our initial strategy may have missed and whether we should start over with a new data freeze. We found several trials with sex comparisons had been added to Trialtrove between April and November 2022. This finding led us to take the second data freeze and to run one consolidated query on that. Additionally, we systematically assessed any papers reporting results on these 75 selected trials to identify gaps in our search strategy and to assess the robustness of our approach. The quality control revealed one syntactic structure we had missed until November that detects some sex comparisons that showed no difference between males and females. We also found that trial results were

sometimes misplaced in the Trial Notes column instead of the expected Trial Results column. These gaps were filled in our final query.

While revising the study, we did a second, similar quality control assessment using all 147 trials with an enrollment [175, 199] selected from the December 23, 2022 data freeze that appeared to have Results. This assessment was completed on September 23, 2023 and did take into consideration any papers about the 147 trials that had been published after the December 23, 2022 data freeze. This assessment did not reveal any sex comparisons that were in Trialtrove but missed by our analysis, so there was no need to revise our analysis methods.

### Analysis of trends over time

To assess the proportion of trials that have a sex comparison as a function of time, we classified each trial according to the year of its start date, which is a structured field in Trialtrove. For sets of consecutive years, we computed the ratio of trials that have sex comparisons according to our curation (numerator) divided by the number of trials that passed our first layer of filters to be candidates to have a sex comparison (denominator). We also partitioned the trials by phase to assess whether trials of different phases had a higher or lower rate of sex comparisons. Only phases II and III had enough sex comparison trials to do a meaningful analysis.

### Computational tools

All automated steps were performed using Python3 (v3.8 or v3.9). For data organization and analysis, the Python package Pandas (v1.4.2) was used. Figures were built using GraphPad Prism (v9.5.1), Matplotlib (v3.5.1), Adobe Photoshop (v13.0), Adobe Illustrator (v28.2), and BioRender.

### Reporting summary

Further information on research design is available in the Nature Portfolio Reporting Summary linked to this article.

## Data availability

The raw data from Trialtrove are available under restricted access only to license holders of Trialtrove via https://citeline.informa.com/trials/results. The processed data for all figure panels that have numerical data are available in the Source Data file. Additional processed data generated in this study are available in the Supplementary Tables and the Supplementary Data files. Source data are provided with this paper.

## Code availability

The most important Python programs used in this study are available via https://github.com/ruppinlab/ProcessTrialtrove and are also available at Zenodo via https://doi.org/10.5281/zenodo.10713794[73]. The programs can be run only if one has a Trialtrove license and can download trial data. Other readers may find the programs useful to read to understand in detail how we processed the Trialtrove data.

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

## Acknowledgements

This research was supported in part by the Intramural Research Program of the NIH, NCI (ER). This research was supported in part by NIH grants 5U01CA180888-08 and 5UG1CA233198-05 (RK). This work utilized the computational resources of the NIH HPC Biowulf cluster. (http://hpc.nih.gov).

## Author contributions

Conceptualization of the study: E.R. Data collection: A.V.K. and A.A.S. Data curation: A.V.K., A.A.S., P.S.R with guidance from R.K. and E.R. Software development: A.V.K,, supervised by A.A.S. Selection of methods for data analysis: A.V.K., A.A.S, P.S.R., R.K., and E.R. Literature search of related work: A.A.S. with guidance from P.S.R. and R.K. Visualization: A.V.K. Writing first draft: A.V.K., A.A.S, and P.S.R. Writing later drafts and editing: A.V.K., A.A.S., P.S.R., R.K., and E.R. Supervised the study: A.A.S. and E.R.

## Funding

## Competing interests

R.K. has received research funding from Boehringer Ingelheim, Debiopharm, Foundation Medicine, Genentech, Grifols, Guardant, Incyte, Konica Minolta, Medimmune, Merck Serono, Omniseq, Pfizer, Sequenom, Takeda, and TopAlliance and from the NCI; as well as consultant and/or speaker fees and/or advisory board/consultant for Actuate Therapeutics, AstraZeneca, Bicara Therapeutics, Inc., Biological Dynamics, Caris, Datar Cancer Genetics, Daiichi, EISAI, EOM Pharmaceuticals, Iylon, LabCorp, Merck, NeoGenomics, Neomed, Pfizer, Prosperdtx, Regeneron, Roche, TD2/Volastra, Turning Point Therapeutics, X-Biotech; has an equity interest in CureMatch Inc. and IDbyDNA; serves on the Board of CureMatch and CureMetrix, and is a co-founder of CureMatch. E.R. is a co-founder of Medaware Ltd, Metabomed Ltd and of Pangea Biomed, Ltd (divested from the latter). E.R. serves as a non-paid scientific consultant to Pangea Biomed, Ltd. The other authors declare that they have no potential competing interests.
