## [Peer Review File · Nature Communications]

Reviewers' Comments:

Reviewer #1:

Remarks to the Author:

In principle the aim of the work is very interesting and covering aspects that have not been investigated in detail as the sex difference in outcomes after the oncological drugs. But some questions arise.

a) It was examined "trialtrove" for a long period and many things changed greatly including law and norms and knowledge on the importance of sex and gender. Therefore, in my opinion, the stratification of this interval in subinterval linked to change in norm and law including FDA norm because the main topic is drug effect and would be of interest to know registration and post-registration trials and if this stratification changes the results.

b) It will be useful to know the phase of clinical studies of pre-registration trials

c) Another issue is the lack of information about the age because it is well known that age influences sex differences being also aging sexual dimorphic process (Nature Cardiovascular Research 1, 844–854 (2022) Elife 2021 May 13;10:e63425, Cell Metabolism 23:1022–1033, and many others). The point of age is of special interest in drug response because it is well known the drug response is age dependent including pharmacokinetics, pharmacodynamic and safety profile (Pharmacological Research 121, 2017, 83-93). For example, globally, old women have a Cmax higher than men (Eur Geriatr Med. 2022 Jun;13(3):559-565), but also other pharmacokinetic parameters changes in a sexual dimorphic manner (Pharmacol Res Perspect 2021 May;9(3):e00775, <https://doi.org/10.3389/fragi.2023.1172789>, Pharmacological Research 121, 2017, 83-93).

d) Dose is a key point for drug response for both efficacy and safety profile, the paper does not report any information about this point which is essential. administered to men and women. Obviously this is a fundamental point

Line 48 please add a more recent references such as Pharmacol Rev 2021 Apr;73(2):730-762. After drug behavior

Line 49 elimination is included in pharmacokinetics

Line 50 please add a more references to highlight that the problem extends to other pathologies and it is not limited to cancer

53-62 It is clear what the authors want to underlie but they should recall that study sex differences is not only a question of recruitment of men and women but there is lot of concerns as illustrated by many authors.

The discussion is quite generic and should discuss properly the role of age and of the dose

Reviewer #2:

Remarks to the Author:

This study addresses an important set of questions-- whether oncology trials report statistical comparisons between sexes and when they do whether they find important differences in outcomes or toxicity. More, its sample is uniquely large, analyzing 89,221 trials in trialtrove. However, the paper is a bit hard to follow, for a few reasons, which are addressable. First, it would be helpful to have a formal methods section between the introduction and results sections of the paper. Currently, some methods are briefly described in the introduction and the results section, but the actual methods section is at the end of the paper. Further there seem to be some new results presented in the discussion section (ie the total number of patients enrolled in the analyzed trials). There are some fragments and incomplete sentences in need of revision, for example, see line 87, which states, "Recent large studies that similarly used Trialtrove include."

In terms of the analyses, it could be helpful to add a sensitivity analysis comparing reporting of statistical comparisons between sexes in clinicaltrials.gov and or publications with what is in trialtrove. As currently structured, the paper reads more about what is curated by trialtrove than more generally in clinicaltrials.gov or the medical literature. If these data exist and trialtrove has not prioritized curating them – that could be an important finding. Without the analysis I suggested I am unsure how representative these findings are of real world practices versus trialtrove curation practices. Overall, an important topic.

Reviewer #3:

Remarks to the Author:

This is a retrospective analysis to explore the outcome differences by sex in oncology clinical trials. Generally, this study highlights an important issue that statistical comparisons between sexes in oncology trials are rarely reported, meaningful differences in outcome and toxicity exist. To some extent, this paper is well-written and has relatively high clinical implications. However, the manuscript can be improved and the following comments need to be adequately addressed:

1. For figure 2-4, the authors indicate the outcome differences of clinical trials through "male", "female", and "same". I cannot think the outcome differences of these trials indicated by the "same" reported the same outcome differences between male and females. In fact, it is that the differences are not statistically significant. So the authors should kindly revise this information.

2. Authors revealed that only 472/89,221 oncology clinical trials 25 (0.5%) had curated post-treatment sex comparisons. Whether this result indicate the sex difference analysis is not mandatory for clinical trial design? If yes, whether the conclusions of this study can be applied for further trials?

3. In Methods, authors should the clarify the update time of ClinicalTrials.gov in Trialrove database. Meanwhile, please provide detailed information when using Python.

4. Authors should provide more in-depth discussion of how this study can impact existing and further clinical trial policies.

Responses to reviewers:

Preliminary remark: *We thank all three reviewers for excellent suggestions that guided us to improve our manuscript. Numerous new references suggested directly and indirectly by the reviewers appear after our responses to the comments from reviewer #3. In this document, the citations are in the format (surname et al., year), which is more informative, while in the revised manuscript, the citations are superscript numbers, which is more compact.*

Reviewer #1 - Role of gender in disease, systematic reviews (Remarks to the Author):

In principle the aim of the work is very interesting and covering aspects that have not been investigated in detail as the sex difference in outcomes after the oncological drugs

Response: *We thank reviewer 1 for the positive assessment and especially for recognizing that our study investigates sex differences in oncology clinical trials in more detail than has been done previously.*

But some questions arise.

a) It was examined “trialtrove” for a long period and many things changed greatly including law and norms and knowledge on the importance of sex and gender. Therefore, in my opinion, the stratification of this interval in subinterval linked to change in norm and law including FDA norm because the main topic is drug effect and would be of interest to know registration and post-registration trials and if this stratification changes the results.

Response: *We agree and stratification by two-year periods was done in Supplementary Figure 1 in our initial submission, but we did not emphasize the findings that the proportion of trials with sex comparisons has declined steadily in the past 15 years or so. The source data for Supplementary Figure S1 have now been added in the new Supplementary Table S4. From Supplementary Figure 1, we see that the proportion of trials with sex comparisons, especially the most important Phase III trials declined since 2002-2003 and especially between 2008-2009 and 2016-2017.*

We rewrote and expanded the pertinent text in Results to read as follows:

The proportion of trials with sex-specific comparisons decreased over time in trials of all phases, and this finding was consistent when trials were further split to Phase II and Phase III only (Supplementary Figure 1, Methods, Supplementary Table S4). The decline was especially substantial between trials with starting years in 2008-2009 (58/2030 trials had sex comparisons) and trials with starting years in 2016-2017 (20/1985 trials). This difference in proportions is highly significant ($P < 4E-5$, Fisher’s exact test, two-sided). We suggest that there are two overlapping reasons for this decline. First, as explained in the Discussion, in the first decade of the 21st century caution among biostatisticians about subgroup

comparisons increased, especially regarding post hoc subgroup comparisons. Second, as we observed in the second paragraph of Results, recent trial papers tend to report separate analysis of males and females as well as other subgroups but tend to avoid direct comparisons between subgroups. For readers who wish to analyze the trends using time intervals different for the two-year intervals we used, we provide more fine-grained data by single year and for each trial phase in Supplementary Table S5. As an influential anecdotal example, we mention a meta-analysis of sex and response to immune checkpoint blockade, all 23 of the underlying trials analyzed males and females separately without a direct comparison (Wallis et al. 2019). Of note, we found the highest fraction of candidate trials with sex comparison in the Phase III set (Supplementary Figure 1), suggesting that as drugs near requests for regulatory approval, doing comparisons by sex increases in importance.

It will be useful to know the phase of clinical studies of pre-registration trials

Response: *Thanks. The combination of phases and starting two-year period was visualized in Supplementary Figure S1 in our initial submission. The source data for Supplementary Figure S1 have now been added in the new Supplementary Table S4. We recognize that our findings in the previous response are unexpected, and readers may want to reanalyze the data using time intervals other than two-year windows. Therefore, we added source data for each combination of (start year, phase) in the new Supplementary Table S5. Both of these new Tables are cross-referenced in the new text in Results, which is copied in our response to the immediately preceding comment from the reviewer.*

b) Another issue is the lack of information about the age because it is well known that age influences sex differences being also aging sexual dimorphic process (Nature Cardiovascular Research 1, 844–854 (2022) Elife 2021 May 13;10:e63425, Cell Metabolism 23:1022–1033, and many others). The point of age is of special interest in drug response because it is well known the drug response is age dependent including pharmacokinetics, pharmacodynamic and safety profile (Pharmacological Research 121, 2017, 83-93). For example, globally, old women have a Cmax higher than men (Eur Geriatr Med. 2022 Jun;13(3):559-565), but also other pharmacokinetic parameters changes in a sexual dimorphic manner (Pharmacol Res Perspect 2021 May;9(3):e00775, <https://doi.org/10.3389/fragi.2023.1172789>, Pharmacological Research 121, 2017, 83-93).

Response: *We thank the reviewer for the idea of including an analysis by age. Regrettably, what we can do with Trialrove data is limited by its nature. We added in Results (page 7) a subsection entitled “Limited information about ages” containing the following comments and analysis:*

Our primary analyses did not consider patient ages, although there are known interactions between sex and age affecting response to treatment (Austad & Fischer 2016; Hägg S. & Jylhävä 2021; Stader & Marzolini 2022; Tannenbaum

et al. 2017; Trenaman et al. 2021; Zhernakova et al. 2022). Trialrove coarsely annotates the ages eligible for a trial as any subset of three values, Children (ages 1-17), Adults (18-64), Older Adults (65-), and 71,832/89,221 have such an annotation. Among the oncology trials with an age annotation, the vast majority 59,792/71,832 (83%) have the age annotation “Adults; Older Adults”. Among the trials with a sex comparison and an age annotation, the proportion annotated as “Adults; Older Adults” is similar at 294/362 (81%). The proportion of all trials annotated as exclusively “Older Adults” is only 1261/59782 (2.1%), precluding any further analysis of significance into these differences.

d) Dose is a key point for drug response for both efficacy and safety profile, the paper does not report any information about this point which is essential. administered to men and women. Obviously this is a fundamental point

Response: *Thanks. The issue of dosage is a very important limitation. We added to the Discussion (page 14)*

Another major limitation of our study is lack of access to individual patient-level data. This hampers our ability to assess potential confounders contributing to our findings on sex-specific differences, such as patient age or total dose of drug received. The interaction effect between sex and age on morbidity and mortality is well established, with women living longer and experiencing greater frailty in older age, but we are not able to observe the effect of this interaction with the available data (Austad & Fischer 2016; Hägg & Jylhävä 2021; Knufinke et al. 2023; Zhernakova et al. 2022). Similarly, pharmacology studies have demonstrated that pharmacokinetics and pharmacodynamics are different between males and females (Mauvais-Jarvis et al. 2021; Moyer et al. 2019; Madla et al. 2021; Trenaman et al. 2021; Soldin et al. 2011; Tannenbaum et al. 2017; Stadler & Marzolini 2022; Knufinke et al. 2023), with these factors as contributors for women experiencing more side effects (potentially older women more specifically). Regrettably, only 33 of the 472 trials in Trialrove that performed sex-specific comparisons describe any data collection related to any of the cytochrome P450 genes involved in drug metabolism, precluding any systematic analysis of such interactions.

Line 48 please add a more recent references such as Pharmacol Rev 2021 Apr;73(2):730-762. After drug behavior

Response: *Thanks for this suggestion. We added citations to (Moyer et al. 2019, Mauvais-Jarvis et al. 2021, Madla et al. 2021, Trenaman et al. 2021) after “drug behavior”; among these, (Mauvais-Jarvis et al. Pharmacol Rev 2021 Apr;73(2):730-762) is the reference that the reviewer specifically suggested.*

Line 49 elimination is included in pharmacokinetics

Response: *Thanks for this correction. We changed the sentence to use the terminology of (Mauvais-Jarvis et al. Pharmacol Rev 2021 Apr;73(2):730-762): “For instance, body composition and metabolism differences between the sexes might influence pharmacokinetics and pharmacodynamics drugs.”*

Line 50 please add a more references to highlight that the problem extends to other pathologies and it is not limited to cancer

Response: *We added citations to (Tannenbaum et al. 2017; Moyer et al. 2019; Trenaman et al. 2021). We changed “including cancer” to ‘including diabetes, cardiovascular diseases, and cancer”*

According to <https://www.nature.com/documents/ncomms-submission-guide.pdf>, “As a guideline, Articles allow up to 70 references” Our initial submission had 56 references and our revised submission has 71 references thanks to the helpful suggestions of reviewer #1

53-62 It is clear what the authors want to underlie but they should recall that study sex differences is not only a question of recruitment of men and women but there is lot of concerns as illustrated by many authors.

Response plan: *We welcome the opportunity and the reviewer’s suggestion to expand this part of the Introduction to list additional concerns and challenges in study sex differences in clinical trials. We replaced the single sentence:*

“To do useful comparisons by sex, requires sufficient clinical trial arm recruitment of both sexes (Mendis et al. 2021) and the need to evaluate treatments in both sexes has regulatory implications (Gispén-de Wied, C. & de Boer 2018).”

With the following paragraph (page 3-4):

Designing, analyzing, and evaluating clinical trials to examine and compare treatments in both sexes requires careful planning. It is important to recruit sufficiently many males and females to have some power to detect differences (Mendis et al. 2021). Regulators must be clear and consistent about what sex-specific subgroup analyses and sex comparison analyses are expected in filings reporting trial results (Gispén-de Wied, C. & de Boer 2018). Differences in proper doses between males and females should be planned and should also consider the patient’s age and pharmacogenomic markers (Mauvais-Jarvis et al. 2021, Stader & Marzolini, 2022; Tannenbaum et al. 2017, Trenaman et al. 2017). Differences in death rates and aging patterns should be considered when comparing survival characteristics of males and females, especially when studying diseases, such as cancer, that predominantly afflict older patients (Dong et al., 2020; Hägg, S. & Jylhävä 2021; Lee et al., 2018; Zhernakova et al. 2022).

The discussion is quite generic and should discuss properly the role of age and of the dose

Response plan: *We revised the Discussion substantially including a paragraph about age and dose, which we copied above, and the following new paragraph about policy implications (pages 14-15):*

Our analysis showcases the impact and room for improvement in current policies to identify sex-specific results in clinical trials. A 2014 US Food and Drug Administration (FDA) Action Plan (<https://www.fda.gov/media/89307/download>) highlighted 27 actions divided into the priorities of "improving completeness/quality of demographic subgroup data collection, reporting, and analysis; identifying barriers to subgroup enrollment in clinical trials and employing strategies to encourage greater participation; and making demographic subgroup data more available and transparent." Additionally, projects awarded by the FDA's Office of Women's Health Research will start to address some of the questions we bring up with this work, but this group has the funding for only a limited number of projects with a duration of 1-2 years per project (<https://www.fda.gov/science-research/womens-health-research/list-owh-research-program-awards-funding-year#2024>). Prioritization cannot be at the level of the FDA alone. Incentives for recruiting sufficient patients and performing these comparisons must also be at the level of journals. As of 2016, several top-tier scientific and oncology-specific journals and journal families, including The Lancet family, Journal of the National Cancer Institute, the Cell family, the Nature family, and the Science family, have adopted SAGER guidelines that require reporting of sex / gender of participants and, to some extent, justification for inadequate powering for subgroup analysis (Heidari et al. 2016; Schiebinger et al. 2016). SAGER guidelines or some similar alternative should be the norm across journals.

Reviewer #2 - gender disparities, systematic reviews (Remarks to the Author):

This study addresses an important set of questions-- whether oncology trials report statistical comparisons between sexes and when they do whether they find important differences in outcomes or toxicity. More, its sample is uniquely large, analyzing 89,221 trials in trialrove.

Response: *We thank reviewer 2 for the positive assessment and especially for recognizing that the questions in our study about sex differences in clinical trials are important.*

However, the paper is a bit hard to follow, for a few reasons, which are addressable. First, it would be helpful to have a formal methods section between the introduction and results sections of the paper. Currently, some methods are briefly described in the introduction and the results section, but the actually methods section is at the end of the paper.

Response: *Thanks. We also share reviewer #2's preference to have the section order as Introduction, Methods, Results, Discussion. However, according to the instructions for Nature*

Communications articles at <https://www.nature.com/ncomms/submit/article>, the required order of sections is Introduction, Results, Discussion, Methods. The reviewer perceived correctly that we gave a short overview of the methods in Introduction and Results aiming to guide readers who are methods-oriented. To address the reviewer's concern given these guidelines, we have now explicitly mentioned in the Introduction (page 5) and Results (page 6) that "methods-oriented readers are encouraged to skip from here to read all of the **Methods** section and then come back to this paragraph."

Further there seem to be some new results presented in the discussion section (ie the total number of patients enrolled in the analyzed trials).

Response: Thanks for pointing this out. Considering your comment, we now moved the information about total enrollment to page 6 of Results.

There are some fragments and incomplete sentences in need of revision, for example, see line 87, which states, "Recent large studies that similarly used Trialrove include."

Response: Thanks. We completed the sentence on line 87, now on page 5, to read:

Recent large studies that similarly used Trialrove include a study predicting drug approvals, a study about the use of germline information in clinical trials, and a catalog of immunotherapy trials.⁴³⁻⁴⁵

We identified some other sentence fragments and sentences that were too long and rewrote them into short full sentences.

In terms of the analyses, it could be helpful to add a sensitivity analysis comparing reporting of statistical comparisons between sexes in clinicaltrials.gov and or publications with what is in trialrove. As currently structured, the paper reads more about what is curated by trialrove than more generally in clinicaltrials.gov or the medical literature. If these data exist and trialrove has not prioritized curating them – that could be an important finding. Without the analysis I suggested I am unsure how representative these findings are of real world practices versus trialrove curation practices. Overall, an important topic.

Response: Thanks. The reviewer raises two important questions of quality control and alternative sources of information.

First, does ClinicalTrials.gov include sex comparisons of outcomes? The answer is very rarely for several reasons. We did additional curation analysis and we added the following to Results (page 6):

In general, Trialrove contains substantial information absent from ClinicalTrials.gov. This is especially true for sex comparisons of results, for which ClinicalTrials.gov contains almost no information. We evaluated on September 12, 2023 all 316 trials for which we found at least one sex comparison with a difference (not necessarily statistically significant). Among these 316 trials: 90 are not in ClinicalTrials.gov at all because they are in registries outside the USA, 142 are in ClinicalTrials.gov without any results, 79 mention sex only

with respect to enrollment, which is called “Participant Flow”, and only 5 mention sex anywhere other than Participant Flow. Only 1 of these (NCT00418886) has separate analysis by sex. Ironically, the clearest indication that ClinicalTrials.gov does not record sex comparisons is trial NCT01274338 in which the associated paper entitled “Enhanced immune activation within the tumor microenvironment and circulation of female high-risk melanoma patients and improved survival with adjuvant CTLA4 blockade compared to males” (Saad et al. 2022) describes the sex comparison in the title; this paper is listed in the publications associated with trial NCT01274338. However, the Results subsection of the ClinicalTrials.gov entry for NCT01274338 has no analysis by sex. We recognize that ClinicalTrials.gov has neither the staff to do the curation nor the enforcement powers to require clinical trialists to deposit their results, so the point of this analysis is to quantify the value added by the Trialrove curation of thousands of sources.

Second, does Trialrove capture most of the sex comparisons available in publications? We did such a sensitivity analysis of 75 trials for the original submission and found that Trialrove curation almost always finds any statistically significant differences in the paper main documents but misses some comparisons that yield non-significant results, especially if they are in the supplementary information. Regrettably, we relegated most of this sensitivity analysis to the Supplementary Information (including what was Supplementary Table 2) in our original submission. We now put more of this information into the main document including converting that Table to what is now Table 1, placed after the References. We substantially enlarged and completely rewrote the main Results text about the sensitivity analysis as follows (pages 7-9).

Our analysis relies on Trialrove curation, which could miss sex comparisons. Therefore, after doing most of our search curation, we selected 75 trials with large enrollments ≥ 200 including both males and females that appeared to have Trialrove-curated results but did not have a sex comparison identified by our analysis (see Supplementary Methods subsection entitled “Quality Control of Query and Preliminary Results”). It is important to understand that these are among the largest oncology trials done and were chosen because the large sizes imply that the trials are likely to have sufficient statistical power to detect a sex difference in outcomes if one exists. We intentionally did this quality control analysis shortly before finishing our curation so that if we found possible improvements in our methods, we could implement those improvements and we did implement one improvement.

To this end, we searched in detail any papers and abstracts published about those 75 large trials to see if any sex comparisons were missed by Trialrove curators. Considering that we found fewer not significant comparisons than significant comparisons (Figure 1b), we expected ahead of time that Trialrove curators may have missed some not significant comparisons. The results of our quality control analysis are in Table 1 (placed after the References).

For only 1 of 75 large trials we checked, there was a statistically significant sex comparison in the main document of the publication that Trialrove curators missed. We infer that Trialrove curators found the large majority of statistically significant published sex comparisons. As expected, there was a larger number of trials (8/75) that had an insignificant (7/75) or marginally significant (1/75, whether it is significant depends on not correcting for multiple tests) sex comparison relegated to the supplementary information; hence, Trialrove curators missed the sex comparisons in those 8/75 publications. For the other 66/75 trials, no sex comparison was published. Importantly, a larger proportion of papers (17/75) had a subgroup analysis by sex in which males and females were analyzed separately but no comparison was done. In the next paragraph and in the Discussion, we hypothesize as to why we think this practice of subgroup analysis of males and females separately with no sex comparison has arisen.

*At the suggestion of a reviewer, we added a second, larger assessment of all 147 trials that were eligible for our main analysis and were not found by us to have a sex comparison curated in Trialrove, appeared to have results, and had enrollments in the slightly smaller range of [175,199]. The enrollment criterion was selected to prefer large trials that have power to find sex differences while avoiding any overlap with the first assessment, which required enrollment ≥ 200 . Encouragingly, we found only 2/147 trials with a significant SOR or side effect sex comparison that Trialrove curators missed and 0/147 that our search methods missed (Table 1). The main differences in the outcomes of the two quality control assessments were an increase in the proportion of studies with a paper but no analysis by sex (51% in the first assessment and 65% in the second assessment) and a corresponding decrease (23% in the first assessment to 7% in the second assessment) in the studies with a paper that did a separate assessment of males and females. Possible reasons for this difference include that i) larger trials are more likely to have power to analyze males and females separately and ii) larger trials are more likely to be published in very high impact journals such as New England Journal of Medicine and Journal of Clinical Oncology, which have developed standardized, in-house figure designs for forest plots that are used to illustrate subgroup analyses, such as separate analyses of males and females. Thus, it appears that authors who publish their trials in the top journals look at how subgroup analyses in their journal of choice were done and imitate previous publications that analyzed males and females separately. The trials with enrollment < 200 are naturally less likely to be published in the highest impact journals and the lower impact journals do not necessarily encourage authors to do analyses by sex, either separately or in comparison (see **Discussion**).*

Reviewer #3 - Cancer clinical trials (Remarks to the Author):

This is a retrospective analysis to explore the outcome differences by sex in oncology clinical trials. Generally, this study highlights an important issue that statistical comparisons between sexes in oncology trials are rarely reported, meaningful differences in outcome and toxicity exist. To some extent, this paper is well-written and has relatively high clinical implications.

Response: *We thank reviewer 3 for the positive assessment and especially for recognizing that the issue of sex differences in clinical trials is important.*

However, the manuscript can be improved and the following comments need to be adequately addressed:

1. For figure 2-4, the authors indicate the outcome differences of clinical trials through “male”, “female”, and “same”. I cannot think the outcome differences of these trials indicated by the “same” reported the same outcome differences between male and females. In fact, it is that the differences are not statistically significant. So the authors should kindly revise this information.

Response: *Thanks. We changed ‘same’ to “not different” in the text and the more compact “no diff.” in the Figures. In the Supplementary Methods, page 5, we modified and added the following text to clarify:*

Additionally, when annotating comparisons for evidence type used, the possible options are Multivariate Analysis, Univariate Analysis, Other Numerical Comparison, No Numerical Comparison. Using these categories, a trial has a comparison that is not different (“no diff.” in the Figures) if either:

A) The evidence type is Multivariate Analysis or Univariate Analysis and the comparison was not statistically significant or

B) The evidence type is Other Numerical Comparison or No Numerical Comparison and the comparison described the comparison with an adjective such as “same” or “similar” or “nearly identical”.

2. Authors revealed that only 472/89,221 oncology clinical trials 25 (0.5%) had curated post-treatment sex comparisons. Whether this result indicate the sex difference analysis is not mandatory for clinical trial design? If yes, whether the conclusions of this study can be applied for further trials?

Response: *Thanks for raising this central and important point. As we explained at the beginning of the Introduction, sex difference analysis is mandated by law in the United States. Sex difference analysis may not be mandated in other countries. One of our key conclusions is the current United States law is not being enforced. Therefore, we added in Results (page 10):*

It may be argued that the denominator of 89,221 is an overestimate for several reasons. Therefore, we counted the subset of trials that had an enrollment of more than 25, enrolled both males and females, had a known start year of 1993-2022, and had at least one data collection location in the United States including Puerto Rico. The last two requirements are because the United States Public Health Service Act sec. 492B, 42 U.S.C. sec. 289a-2 (quoted at the start of the

Introduction) was enacted only in the United States in 1993. The number of trials in the numerator and denominator meeting the four requirements above are 215/17988 (1.2%). The numbers for each start year are shown in Supplementary Table S6.

We moved the previously done analysis over time (Supplementary Figure 1) to follow the above new paragraph. In general, we see no improvement in more recent years. One of our key conclusions is that in the United States the current law and NIH policy are not being enforced. We also noted (page 10) that:

To summarize the analysis over time, we do not observe any increase in the proportion of United States -based trials with a sex comparison after any of four key events: enactment in 1993 of the law mentioned above, establishment of ClinicalTrials.gov in 2000; requirement of trial registration in ClinicalTrials.gov starting around 2007, and implementation of the NIH policy on sex as a biological variable around 2016.

3. In Methods, authors should clarify the update time of ClinicalTrials.gov in Trialrove database. Meanwhile, please provide detailed information when using Python.

Response: *Thanks. We added on page 16:*

Trialrove is updated every weekday and in our experience, new data in ClinicalTrials.gov appear in Trialrove within days or a few weeks. To have a clear and consistent reference point, we had to take a single data freeze (in December 2022) to have a stable set of data to analyze.

We now make available via GitHub two python programs to process Trialrove data. One program formats the downloaded Excel files into text files that are more easily searched for some purposes. The other program was used to find the initial set of 4,061 candidate trials that may have a sex comparison. This is mentioned in Code Availability on page 19.

4. Authors should provide more in-depth discussion of how this study can impact existing and further clinical trial policies.

Response plan: *We have substantially rewritten the **Discussion**, including mentioning the issues of age and dose. Regarding policies, we added this paragraph (pages 14-15):*

Our analysis showcases the impact from and room for improvement in current policies to identify sex-specific results in clinical trials. A 2014 US Food and Drug Administration (FDA) Action Plan (<https://www.fda.gov/media/89307/download> highlighted 27 actions divided into the priorities of "improving completeness/quality of demographic subgroup data collection, reporting, and analysis; identifying barriers to subgroup enrollment in clinical trials and employing strategies to encourage greater participation; and making demographic subgroup data more available and transparent." Additionally, projects awarded by the FDA's Office of Women's Health Research

will start to address some of the questions we bring up with this work, but this group has the funding for only a limited number of projects with a duration of 1-2 years per project (<https://www.fda.gov/science-research/womens-health-research/list-owh-research-program-awards-funding-year#2024>). Prioritization cannot be at the level of the FDA alone. Incentives for recruiting sufficient patients and performing these comparisons must also be at the level of journals. As of 2016, several top-tier scientific and oncology-specific journals and journal families, including The Lancet family, Journal of the National Cancer Institute, the Cell family, the Nature family, and the Science family, have adopted SAGER guidelines that require reporting of sex / gender of participants and, to some extent, justification for inadequate powering for subgroup analysis. (Heidari et al. 2016; Schiebinger et al. 2016). SAGER guidelines or some similar alternative should be the norm across journals.

References [All but one of the following references are new in our revision and are cited in response to comments from the reviewers and are also cited by number in the revised manuscript. The paper by Wallis et al. is the exception, which was cited in our original submission and is cited in one response above.]

Austad, S.N. & Fischer, K.E. Sex differences in lifespan. *Cell Metab.* **23**, 1022-1033 (2016). (reference 47 in the revised manuscript)

Hägg, S. & Jylhävä, J. Sex differences in biological aging with a focus on human studies. *eLife.* **10**, e63425 (2021). (reference 21 in the revised manuscript)

Heidari, S., Babor, T., De Castro, P., Tort, S. & Curno M. Sex and Gender Equity in Research: rationale for the SAGER guidelines and recommended use. *Res. Integr Peer Rev.* **1**, 2 (2016). (reference 63 in the revised manuscript)

Knufinke, M., MacArthur, M.R., Ewald, C.Y. & Mitchell, S.J. Sex differences in pharmacological interventions and their effects on lifespan and healthspan outcomes: a systematic review. *Front Aging.* **4**, 1172789 (2023). (reference 62 in the revised manuscript)

Madla, C.M., et al. Let's talk about sex: Differences in drug therapy in males and females. *Adv Drug Deliv Rev.* **11**, 3804 (2021). (reference 6 in the revised manuscript)

Mauvais-Jarvis, F., et al. Sex- and gender-based pharmacological response to drugs. *Pharmacol Rev.* **73**, 730-762 (2021). (reference 4 in the revised manuscript)

Moyer, A.M., Matey, E.T. & Miller V.M. Individualized medicine: sex, hormones, genetics, and adverse drug reactions. *Pharmacol Res Perspect.* **7**, e00541 (2019). (reference 5 in the revised manuscript)

Saad, M., et al. Enhanced immune activation within the tumor microenvironment and circulation of female high-risk melanoma patients and improved survival with adjuvant CTLA4 blockade compared to males. *J Transl Med.* **20**, 253 (2022). (reference 46 in the revised manuscript)

Schiebinger, L., Leopold, S.S. & Miller VM. Editorial policies for sex and gender analysis. *Lancet* **388**, 2841–2842 (2016). (reference **64** in the revised manuscript)

Stader, F. & Marzolini, C. Sex-related pharmacokinetic differences with aging. *Eur Geriatr Med.* **13**, 559-565 (2022). (reference **19** in the revised manuscript)

Tannenbaum, C., Day, D. & Matera Alliance. Age and sex in drug development and testing for adults. *Pharmacol. Res.* **121**, 83-93 (2017). (reference **11** in the revised manuscript)

Trenaman, S.C., Bowles, S.K., Andrew, M.K & Goralski, K. The role of sex, age and genetic polymorphisms of CYP enzymes on the pharmacokinetics of anticholinergic drugs. *Pharmacol Res Perspect.* **9**, e0775 (2021). (reference **8** in the revised manuscript)

Wallis, C.J.D., et al. Association of patient sex with efficacy of immune checkpoint inhibitors and overall survival in advanced cancers: A systematic review and meta-analysis. *JAMA Oncology* **5**, 529-536 (2019). (reference **31** in the revised manuscript)

Zhernakova, D. & Sinha, T., Andreu-Sánchez, et al. Age-dependent sex differences in cardiometabolic risk factors. *Nat Cardiovasc Res.* **1**, 844-854 (2022). (reference **23** in the revised manuscript)

Reviewers' Comments:

Reviewer #1:

Remarks to the Author:

After the revision the manuscript includes the previous comments and suggestions given more complete information on important topics

Reviewer #3:

Remarks to the Author:

Authors had answered my questions.

Reviewer #4:

None

Reviewer #5:

Remarks to the Author:

The authors have thoroughly, and to the best of their ability, given the data limitations, addressed all reviewer concerns. Limitations are now clearly described, and the manuscript has improved based on the recommended changes.